

# Sandbag Replacement Systems - Stability, Functionality and Handling

Lena Lankenau[1], Christopher Massolle[1], Bärbel Koppe[1], Veronique Krull[1]

[1]Institute for Hydraulic and Coastal Engineering, Hochschule Bremen – City University of Applied Sciences, Bremen,
28199, Germany

*Correspondence to*: Lena Lankenau (lena.lankenau@hs-bremen.de)

**Abstract.** The classic aid in operative flood defence is the sandbag. Over the past few decades, though, so-called sandbag replacement systems (SBRS) have also been available for flood fighting. Although the use of sandbags is time-consuming as well as highly intensive in terms of materials and personnel, so far SBRS are rarely used in Germany. However, owing to their functionality and their labour and time-saving characteristics, they can make an essential contribution to flood protection—and this all the more so in view of the expected consequences of climate change. In order to foster confidence in such systems, the Institute of Hydraulic Engineering at the Hochschule Bremen - City University of Applied Sciences (IWA) carried out a series of systematic tests of SBRS that focused on the functionality, stability and handling of the systems. The experience gained shows that SBRS have the potential to make flood defence more efficient than the use of sandbags alone. Since SBRS are technical systems whose functional capability must be proven before they can be used, it is recommended to introduce an official test and certification procedure.

## 1 Introduction

The classic aid in operative flood defence is the sandbag. So-called sandbag replacement systems (SBRS) have also been available for some time now, although their use is still very limited. Figure 1 shows such mobile flood defence systems: they can be subdivided into tube, basin, flap, trestle, dam or panel systems and bulk elements. The systems protect against flooding either by their bulk weight (water, sand, concrete) or their geometry in connection with the vertical hydrostatic water pressure.

Sandbagging is time-consuming as well as highly intensive in respect of materials and personnel. However, the advantage of using sandbags lies in the possibilities for flexible deployment and many years of practical experience. So far, SBRS are rarely used in Germany. During the Elbe flood in 2013, for example, mobile SBRS were only used sporadically (cf. AQUARIWA, 2019; Mobildeich, 2019), and this despite the fact that they hold the potential for a much more efficient flood defence, as their use entails significantly lower material, personnel and time requirements than conventional sandbagging. The main disadvantage of SBRS is the higher cost of acquisition. However, in contrast to sandbags, SBRS are reusable, do



not have to be disposed of at high cost after a flood event and can be set up and dismantled with considerably less manpower, and the higher acquisition costs can be amortized over subsequent operations.

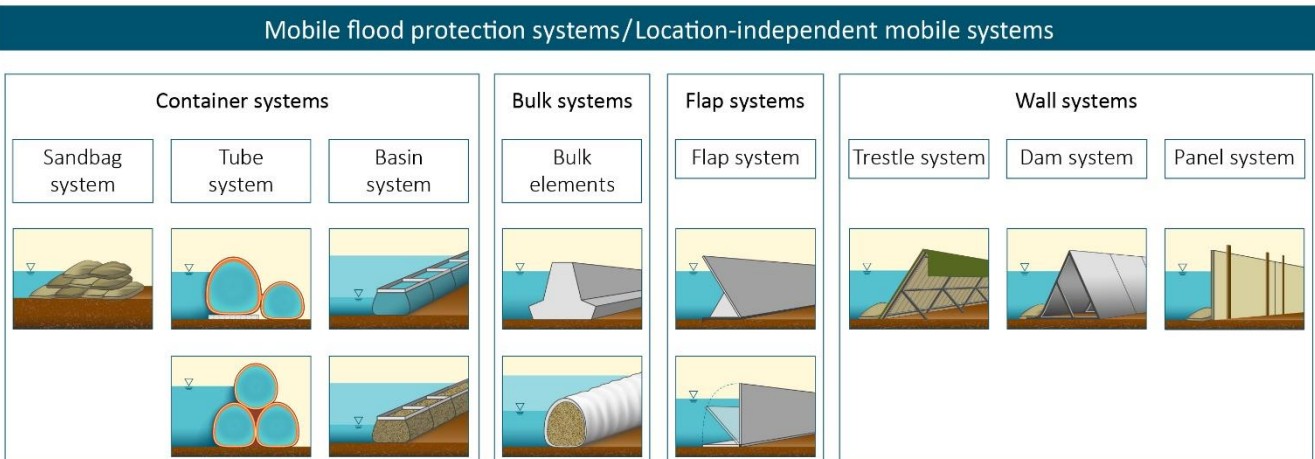

**Figure 1. Classification of mobile, non-location dependent flood protection systems (Massolle et al., 2018).**

In Germany, operational flood defence is regulated as part of hazard prevention or disaster control at the federal state level. Direct responsibility lies at the municipal level and thus with the local districts and cities. This includes the responsibility to provide the necessary material for the protection of the general public, whereby as a rule sandbags — which are the significantly cheaper option — are favoured vis-à-vis SBRS with their higher investment costs. In case of a disaster event, assistance can be requested from the federal state or the federal government, whereby the financing of such assistance will
still remain initially with the affected administrative districts or cities. Ultimately, the costs of major damage events, such as caused by the Elbe floods of 2002, 2006 and 2013, will be borne predominantly by the federal state and the federal government. Once such an event occurs, however, no time can be lost in procuring SBRS, if they are not already standing by. Thus, the cost of procuring and stocking SBRS—in addition to a lack of confidence or knowledge about their functionality—presents a major hurdle to their use.

In order to increase the confidence of decision-makers in SBRS and to promote their use in operational flood defence, it is desirable to carry out systematic tests regarding their suitability for practical use and to develop a procedure for certifying such systems. At the international level, corresponding certification already exists. It can be awarded by the globally active testing and certification service FM Approvals (FM Approvals 2019), based on the American National Standard for Flood Abatement Equipment (ANSI and FM Approvals, 2014), and the British Standard Institution (BSI, 2019a), which is based
on the Publicly Available Specification (PAS) for flood protection products—Specifications Part 2: Temporary Products (BSI, 2014). In Germany, no corresponding certification or testing system for SBRS is currently available. However, there is some information available on the design and the scheduled as well as unscheduled use of SBRS in German-speaking countries: this is currently contained in the recommendations of the leaflet 'Mobile Flood defence Systems' issued by the German Association of Engineers for Water Management, Waste Management and Cultural Construction. (BWK, 2005), the



handbook 'Mobile Flood Protection' of the Austrian Water and Waste Management Association (ÖWAV, 2013) and the decision-making aid 'Mobile Flood Protection' of the Swiss Association of Cantonal Fire Insurers (VKF) as well as the Swiss Federal Office for Water and Geology (BWG) (Egli, 2004).

SBRS can make an essential contribution to operational flood defence owing to their functionality and time-saving
characteristics as well as lower requirements for materials and personnel, and this even more so in view of the expected consequences of climate change. It was therefore decided to carry out systematic testing of SBRS in the test facility of the Institute of Hydraulic Engineering at Bremen University of Applied Sciences (IWA). The focus of the test setups was on functionality and stability as well as handling of the systems. First results of the test setups with regard to installation times, water heads and seepage rates have already been published (Massolle et al., 2018). This article summarises the experience
gained from the test setups with regard to functionality, stability and handling of the systems in accordance with the guidelines for loss prevention of the German insurers for mobile flood defence systems (VdS, 2014), which are in turn based on the recommendations of the BWK (BWK, 2005), the VKF and the BWG (Egli, 2004). The system assessments obtained in this way serve to provide a practical assessment of the operational capability of SBRS.

Comparable system assessments were carried out by the UK Environment Agency (EA) on the basis of three sources of
information; namely, the literature, user workshops with users of systems and interviews with manufacturers and distributors of products. It was found that most of the systems provided adequate protection, but that in some cases operational processes or inaccurate hydraulic assessments led to system failure. The assessments covered the physical, operational and structural characteristics of temporary flood products available on the UK market in 2009. The systems were subdivided into tubular systems, containers, freestanding barriers and frame barriers. Four of the tested systems (NOAQ Tubewall (Öko-Tec
Schlauchwall), Tiger Dam, NOAQ Boxwall, Geodesign (Aqua Barrier)) overlap with the systems shown here. The NOAQ Tubewall is comparable with the Öko-Tec Tubewall and Geodesign as a pallet variant with Aqua Barrier. (Ogunyoye et al., 2011)

In the frame of a Canadian study, in which the authors assessed the suitability of innovative systems as an alternative to sandbags primarily on the basis of the literature, commercial brochures, theoretical considerations and stability calculations,
four different system types were examined. Among the types studied were water-filled systems, inflatable tubular systems, gabion-like systems filled with sand or earth, dam beams and highway barriers. Three of the water-filled systems or inflatable tubular systems are comparable with systems examined in the present study (Aqua Barrier with Hydrobaffle, Clement with Tiger Dam, NOAQ Tubewall with Öko-Tec Schlauchwall); the other system types were not tested. (Biggar and Masala, 1998)

In a study conducted by the University of Kentucky (Mc Cormack et al., 2018), the possible uses of sand-filled temporary flood defence barriers to protect roads from flooding were analysed on the basis of existing operational experience. However, the systems considered are not comparable with those covered in the present study.

For an overview of the SBRS tests performed, the reader can refer to Massolle et al. (2018). Individual systems certified by BSI Kitemark can be found under BSI (2019b) and systems certified by FM Approvals under NFBTCP (2019).



## 2 Description of tests

The tests were carried out in the IWA test facility, which was set up on the premises of the THW Training Centre Hoya as part of the research and development project DeichSCHUTZ (2014 - 2017) for the development of systems to reduce buoyancy in dykes at risk of failure, funded by the Federal Ministry of Education and Research (Massolle et al., 2018). The
5     facility consists of a U-shaped basin, the 15-metre wide opening of which is closed by a dam. For the SBRS tests, various systems were set up across the entire width of the basin parallel to the dam line and the space between the dam and the system was then filled with water (Figure 2). This allows a realistic simulation of the hydrostatic load on the systems. Other possible load parameters such as current, waves, wind, flotsam and vessel impact cannot be investigated in the test facility.

**Flutschutz-Doppelkammerschlauch (T)** | **Hydrobaffle (T)**

**Mobildeich (T)** | **Öko-Tec (T)**

**Tiger Dam (T)** | **AQUARIWA (B)**

**INDUTAINER (B)** | **aqua defence (TR)**

**NOAQ Boxwall (D)** | **Quick Damm M (B), Aquariwa (B), Aqua Barrier (TR)**

10     **Figure 2. The various SBRS tested, (T) Tube system, (B) Basin system, (D) Dam system, (TR) Trestle system.**



At least one of the container types and wall systems shown in Figure 1 was selected for each of the test setups. Flap systems could not be tested because no manufacturer could be found who was prepared to make a suitable system available. Bulk elements and panel systems were not considered. This is because in operational practice the use of bulk elements requires technical aids being available at short notice to install the elements, which is often impractical for logistical reasons or for

reasons of the load-bearing capacity of the foundation soil being impaired during flooding. The use of panel systems is limited to suitable soils and low water levels. Bulk elements and panel systems were therefore not taken into account in the test setups owing to their necessity for framework conditions such as accessibility with heavy equipment and the avoidance of damage to test setups from deep ramming of retaining stakes.

In cases where the suppliers offered more than one system size, a variant suitable for a water head of 0.6 metres was selected

for our test setups. This height corresponds to the recommendations contained in the leaflet 'Mobile Flood Protection Systems' (BWK, 2005) for the unscheduled use of SBRS in operational flood fighting. The recommendation results, on the one hand, from the increasing danger of foundation-surface failure with increasing water levels as well as from not being able to dimension the systems in advance to cope with the loads occurring at an unknown location. Not exceeding the specified maximum water level minimises the risk of damage. If larger system heights are required, the risk must be weighed

on a case-by-case basis. Even if the conditions to be expected could be examined on site by a competent person, if possible prior to the use of an SBRS, the time and information required for this is usually not available. Since some systems are not specifically designed for water heads of 0.6 m, over dimensioned systems such as AQUARIWA, aqua defence, Hydrobaffle, Tiger Dam were used.

The SBRS tested are only a selection of the systems available on the market. In addition, one of the systems investigated, the

Quick Dam Type M, is no longer produced, but still in use. Market analysis showed that some system types, such as basin systems and tube systems, are more frequently present on the market than others. However, the number of products of a system type does not allow conclusions to be drawn about its functionality.

Tube systems and basin systems are usually filled with water to ensure their stability. Not many tube systems or basin systems can be filled with sand. Sand fillings were not considered during the test setups as the requirements for filling and

dismantling could not be met in the test facility. Therefore only tube and basin systems filled with water were tested. The Öko-Tec tube wall is an exception. With this system, the tube is inflated with air. The system is stabilised by a plastic sheet called 'skirt' spread out on the water side of the system, which is friction-locked to the tube. The tube is stabilised solely by the vertical hydrostatic pressure acting on the horizontally laid skirt. No other of the tested systems using an upstream skirt are connected to the system in such a friction-locking manner. An upstream skirt must always be weighted down at the

water-side edge, often with sandbags. The trestle and dam systems do not require filling.



The systems were initially dammed up to a water height of 0.6 metres, in accordance with the recommendations of the BWK leaflet 'Mobile flood protection systems'. After setting a constant seepage rate at a dam height of 0.6 m (cf. Massolle et al. 2018), the water head was further increased in stages until a system failure occurred due to the water volume exceeding the load limits of the system or a partial overflow of the system occurred. The Quick Dam Type M and Aqua Barrier systems

5 were not available in sufficient length and were therefore installed in combination with the AQUARIWA system. The test basin was only briefly filled with water up to a height of 0.6 m. The NOAQ Boxwall system only has a feasible protection height of 0.5 metres, but was nevertheless tested because of its simplicity and speed of installation. In principle, the manufacturer recommends the use of the NOAQ Boxwall System on paved surfaces, as this results in a better sealing effect on the underlying surface. According to the manufacturer's training video, the Tiger Dam system can be used with and without anchoring to the ground or additional plastic skirts on the water side, but is only FM-approvals certified if the skirt and the anchoring system are in place (NFBTCP, 2019). Both variants were investigated. The tightening belts pulled around the tubes were fastened in the area of every second wedge with a rope affixed by stakes on the land side and water side. Finally, a plastic skirt was spread in front of the system on the water side, which reached up to the apex of the upper tube.

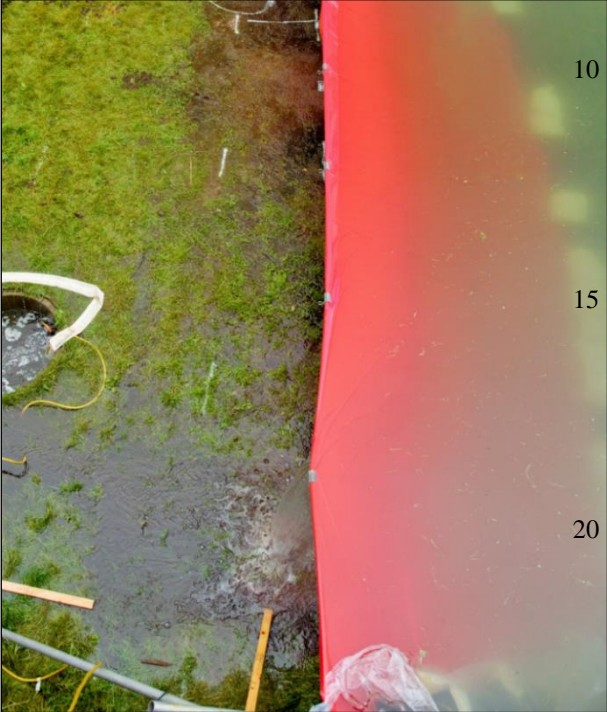

**Figure 3: Overflowing SBRS (aqua defence) (Massolle et al., 2018).**

Full impoundment of the tested systems and water overflow cannot be realized over the entire length of the SBRS due to unevenness of the basin floor and limited pumping capacity. This restriction is particularly relevant in case of occurrence of an overflow load, as the unevenness meant that only a slight overflow height could be achieved in the right-hand area of the test facility (Figure 3).

If overflow occurs when using SBRS, it must be prevented from washing away the soil on the landside, otherwise system failure can occur. The overflowing water must therefore be discharged or distributed over a sufficiently large area. Theoretically, an SBRS can overflow if the system is sealed via vertical water pressure, since with increasing water levels the system is increasingly held stable via the vertical pressure. A protruding skirt on the water-side will afford more

30 protection, as the buoyancy forces under the system are thereby minimised. Whether the system will overflow depends on its geometry and/or bulk. With increasing water levels, the probability of failure due to tilting, slipping or rolling increases. Systems that do not benefit from the effect of vertical water pressure for stabilisation are not stabilised further with an increasing water level. In terms of stability, a high bulk and/or a low centre of gravity are fundamentally advantageous here.





The tests do not take into account the possibility of the foundation soil giving way with increasing water levels, since damming within the test setups only took place on a defined and stable floor.

## 3 Test results

The Guidelines for Loss Prevention issued by German Insurers for Mobile Flood Protection Systems contain a specimen

evaluation form for SBRS, which is intended to serve as a decision-making aid for system evaluation for persons responsible for flood defence (VdS, 2014). The systems tested were evaluated in accordance with these guidelines (Table 1). The evaluation criteria relate to the area of application, stability, procurement and durability, installation, dismantling and maintenance as well as the logistics surrounding the systems. If a specification could not be determined or derived from the results of the test setups, manufacturers' specifications were used, or the evaluation was carried out on the basis of authors'

considerations. The failure mechanisms affecting the surface an SBRS is installed on, such as caused by hydraulic heave or erosion, were not considered due to their dependence on the variable site conditions encountered in operational practice. Also not taken into consideration were the system connections to walls or the like, the possibility of laying the system in curves or with angles or the system behaviour on different substrates (soft, solid, rough, smooth, even, uneven, permeable, impermeable etc.). The evaluation criteria on which the system evaluations are based are described in Table 2.

**Table 1. System Evaluation; DKS: Doppelkammerschlauch (double – chamber  tube)  TD: Tiger Dam, A: Skirt and Anchoring**

| | AQUARIWA | INDUTAINER | Quick Damm | Aqua Barrier | aqua defence | NOAQ Boxwall | Flutschutz-DKS | Hydrobaffle | Mobildeich | Öko-Tec | TD with A. | TD without A. | Explanation / Remarks |
|---|---|---|---|---|---|---|---|---|---|---|---|---|---|
| **Application area** | | | | | | | | | | | | | |
| Uneven ground | - | - | o | o | o | o | o | + | + | o | + | + | |
| Unsurfaced ground | - | - | + | - | - | o | + | + | + | + | o | o | |
| Height of retained water (h) | o* | o | o | o | o | - | o | +* | +* | o* | - | +* | *Manufacturer's data Not all h tested |
| Height adjustable | - | - | - | - | - | - | - | o | - | o | o | |
| Overflowable | o* | - | n/s | o | o | + | - | - | + | + | - | + | *Perchance with sand filling |
| Installation in water | o | - | - | o | o | o | - | +* | +* | - | - | - | * Manufacturer's data |
| Space requirement in use | - | - | o | - | - | + | - | o | - | - | + | - | |
| **Stability** | | | | | | | | | | | | | |
| Tipping stability | - | - | o | o | o | o | + | + | + | + | o | + | |
| Roll / slide stability | + | o | o | + | + | o | o | - | + | o | - | + | |
| Buoyancy stability | + | + | o | + | + | o | o | - | + | o | - | + | |
| Anchoring | - | - | - | o | o | - | - | - | - | + | + | n/s | |
| Resistance against mechanical effects | o | - | o | o | o | - | o | - | + | - | - | o | |
| Resistance against vandalism | - | + | - | - | - | - | - | - | - | - | - | - | |
| Domino effect | + | - | + | o | o | - | - | o | - | - | - | - | |
| **Procurement and durability** | | | | | | | | | | | | | |
| Costs | o | + | n/s | o | - | o | o | o | o | - | + | + | |
| Service life | o/+ | -* | n/s | n/s | n/s | o** | + | + | o** | o** | + | + | *During continuous operation |



| | AQUARIWA | INDUTAINER | Quick Damm | Aqua Barrier | aqua defence | NOAQ Boxwall | Flutschutz-DKS | Hydrobaffle | Mobildeich | Öko-Tec | TD with A. | TD without A. | Explanation / Remarks |
|---|---|---|---|---|---|---|---|---|---|---|---|---|---|
| | *** | | | | | | | | | | | | **Legal warranty<br>***o: Water sack<br>+: GRP panel |
| Reusability | o | o | + | + | + | + | + | + | + | + | + | + | |
| **Installation** | | | | | | | | | | | | | |
| Installation time | o* | o* | n/s | + | + | + | o* | o* | o* | + | -* | -* | *According to pumping capacity |
| Equipment requirement | - | - | o | o | o | + | o | o | o | o | o | - | |
| Persons | + | + | + | + | + | + | + | + | + | + | + | + | |
| Requirement of filling material | o* | o | o* | + | + | + | - | o | o | + | o | o | * Sand filling |
| Number of individual elements | - | o | + | + | o | + | + | + | o | o | - | - | |
| Simplicity of installation | + | + | + | + | o | + | + | + | + | o | - | - | |
| Weight of individual elements | + | + | o | + | + | + | o-* | o-* | o** | o | o | o | * According to system length<br>**With reel |
| **Dismantling and maintenance** | | | | | | | | | | | | | |
| Simplicity of dismantling | o* | + | o* | + | + | + | o | + | + | + | + | + | * Sand filling |
| Disposal costs | o* | o | o* | - | o | + | + | + | + | + | + | o | * Sand filling |
| Cleaning costs | o | - | o | o | o | o | o | o | o | o | o | o | |
| Repairs and spares | + | - | + | + | + | - | + | + | + | + | + | + | |
| **Logistics** | | | | | | | | | | | | | |
| Space for storage/ transport | + | + | o | + | + | + | o | o | o | o | + | + | |

| **Legend** | + = good | o = medium | - = bad | n/s = not specified |
|---|---|---|---|---|

**Table 2. Evaluation criteria.**

| | Evaluation criteria |
|---|---|
| **Area of application** | |
| Uneven ground | Unevenness, curbstones, etc. |
| Unsurfaced ground | Special requirements for the condition of the foundation surface |
| Height of retained water | Height of retained water h up to 0.6 m = -; up to 1.5 m = o; up to 3.0 m = +<br>Observe recommendations for unscheduled use of SBRS according to BWK (2005) |
| Height adjustable | Subsequent increase possible |
| Overflowable | Overflow capability according to manufacturer (M) or determination in authors' tests (AT)<br>No = -; Yes (AT or M) = o; Yes (AT and M) = + |
| Installation in water | Manufacturer's specification or own estimate based on system characteristics |
| Space requirement in use | Depth incl. any upstream skirt ≤1,0 m = +; ≤2,0 m = o; >2,0 m = - (refers to the system variants tested) |
| **Stability** | |
| Tipping stability | Tube systems are less prone to tipping than dam or trestle systems. The heavier the installed systems, the less prone they are to tipping. (Selective) Sinking into the ground increases the risk of tipping. Anchoring or securing against buoyancy counteracts tipping. |
| Roll / slide stability | Tube systems are generally more susceptible to rolling away. The lower the weight and the smoother the foundation surface of the system, the easier it is for the system to slip. Anchoring or securing against buoyancy counteracts sliding or rolling. |
| Buoyancy stability | The risk of system failure due to buoyancy is greater for filled systems with a lower weight. Depending on the shape, buoyancy forces can also act on the water side (e.g. tube systems). Systems with a large |



| | Evaluation criteria |
|---|---|
| | foundation surface which achieve their load bearing effect through the vertical water pressure from the outside also have a greater risk of failure due to buoyancy. An upstream skirt, drainage, seal or anchoring counteracts failure caused by buoyancy. |
| Anchoring | System can be anchored against wind, current, slipping or rolling |
| Resistance to mechanical effects | Susceptibility to damage e.g. by flotsam impact |
| Resistance against vandalism. | Susceptibility to deliberate damage |
| Domino effect | Threat to the entire dam due to failure of individual elements |
| **Procurement and durability** | |
| Costs | ≤100 €/m = +; ≤300 €/m = o; >300 €/m = - (refers to the system variants tested) |
| Service life | Service life according to manufacturer ≤1 year = -; ≤5 years = o; >5 years = + |
| Reusability | Manufacturer's data |
| **Installation** | |
| Installation time | Installation time according to manufacturer or from own test. For all water-filled systems, the installation time depends on the pump used. |
| Equipment requirement | Tarpaulins, sandbags, hoses, pumps, adapters or blowers<br>Tarpaulin and etc. = -; Tarpaulin or etc. = o; no equipment requirement = + |
| Persons | ≤2 Persons = + |
| Requirement of filling material | Sand filling = - ; water filling = o; no filling = + |
| Number of individual elements | Number of individual parts |
| Simplicity of installation | System installation easy to understand and to perform |
| Weight of individual elements | ≤35 kg = +; ≤100 kg = o; >100 kg = - (refers to the tested system variants) |
| **Dismantling and maintenance** | |
| Simplicity of dismantling | System dismantling easy to understand and easy to perform |
| Disposal costs | Foils, tarpaulins, sandbags - Disposal after use |
| Cleaning costs | Effort involved in system cleaning |
| Repairs and spares | Minor damage can be repaired by the user. Material and spare parts are available. |
| **Logistics** | |
| Space for storage/ transport | Compactness of the dismantled system |

The systems were tested on a grass surface and were set up by two people. In some cases, there were major differences between the manufacturer's time specifications and the times measured during the test setups (cf. Massolle et al, 2018). To be set up, the systems had to be transported manually from the edge of the basin to the point of installation and thus over a maximum distance of 15–20 metres. It is quite conceivable that faster installation times can be achieved on surfaces suitable for vehicles to travel on and which offer better logistical conditions. On the other hand, significantly longer manual transport distances — and thus longer assembly times compared to the test conditions — may occur in practice. The installation times for the water-filled SBRS also depend strongly on the available pump capacity and the water supply. In principle, however, it can be said that installation and dismantling of the systems is generally possible with just two persons and is many times faster than the construction of a sandbag dam. In addition, it is also possible to optimise installation times by using more helpers. Systems that have no need of filling also show a clear time advantage during assembly and dismantling.

Setting up the systems is often self-explanatory and instructions are easy to follow. It is still recommended, though, to involve an expert in order to avoid possible assembly errors with their far-reaching consequences. Assembly errors are


always possible. With the Öko-Tec tube system, for example, there is a risk that the drainage mat located under the upstream skirt will be inverted, thus endangering the functionality of the system.

Taking precautions against buoyancy can be generally recommended. Systems such as NOAQ Boxwall, Tiger Dam or Öko-Tec are dependent on this safety precaution. Protection can be ensured by an upstream skirt, a drainage system, a seal on the
water-side edge or anchoring of the system. Systems such as the FLUTSCHUTZ Doppelkammerschlauch (DKS - double-chamber tube) have good protection against failure owing to buoyancy as result of their high bulk weight, and no further measures are called for. However, completely weighting down an upstream skirt with sandbags or other weights is still generally recommended, as this can also considerably minimise the occurrence of seepage (cf. Massolle et al, 2018).

Especially systems with a restricted contact surface are prone to the danger of sinking into saturated ground (aqua defence,
Aqua Barrier, Tiger Dam). This also applies to the AQUARIWA system, the filled base of which is flat, but whose plastic skin lies somewhat unevenly. Precise data on how long it would take for the system to fail due to sinking at the contact surfaces cannot be derived from the test carried out due to its relatively short duration of just a few hours (cf. Massolle et al, 2018). In principle, there is a correlation between the depth of subsidence, the magnitude of the load exerted, the type and the antecedent wetness of the ground underneath as well as the duration of a flood event, which can last up to several days
and even weeks. Some subsidence of the systems lying on a restricted contact surface could be observed during water impoundment, but this did not lead to failure during the test setups, presumably due to the short damming time of just a few

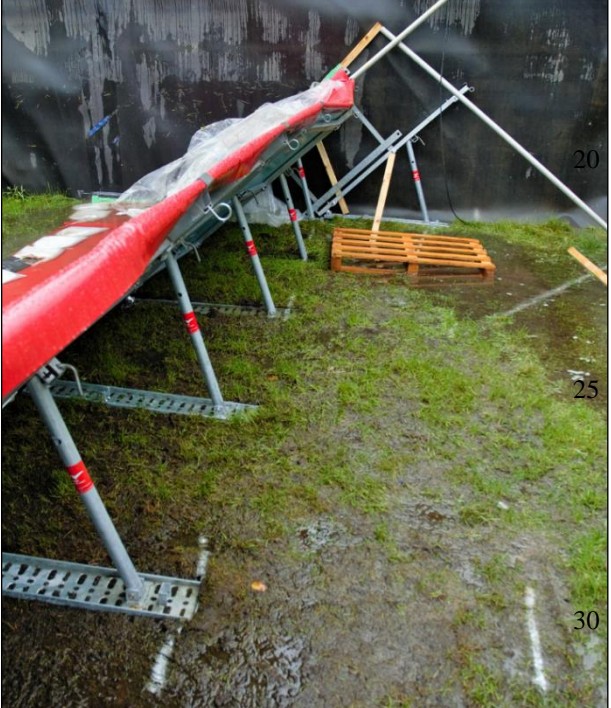

hours. Figure 4 shows the aqua defence system during dismantling. The system sank the deepest into the foundation soil in the area of the greatest water depths during damming—at the top of the picture. In this area, however, the system also overflowed while the test basin was being filled with water, so that some of the increased subsidence was probably due to erosion of the foundation soil.

Especially in the case of fine sandy soils, there is a risk of foundation soil failure due to hydraulic heave or erosion caused by water flowing under the system. Especially when additional pumping is used, care must be taken that the soil under the systems is not removed with the flow of water being pumped out. There is also a risk that the friction between soil and system on paved ground will be reduced by the presence of loose grains of sand or gravel. Here, it is recommended to sweep the areas around the contact surfaces prior to installation. Minor

**Figure 4: Supporting columns sunk into the saturated foundation soil while damming (aqua defence).**

unevenness can be levelled out with sandbags or lime that swells



in contact with water. When installing the systems, attention must be paid to whether there are gradients in the terrain across or along the planned system line, as this would increase the risk of tipping, sliding or rolling. Some systems shifted or were deformed when the test basin was being filled with water, owing to play in their construction or expansion of the material they are made of, but then stabilised again (FLUTSCHUTZ-DKS, Hydrobaffle, Tiger dam, Aqua Barrier). The pending

failure of all the tested systems when overloaded was always indicated by visible shifting, but this was usually so quick that there was no possibility of taking countermeasures over longer lengths.

In summary, it can be stated that all the systems tested remained stable at the water levels specified by their manufacturers. The systems aqua defence, NOAQ Boxwall, Mobildeich, Öko-Tec as well as Tiger Dam with anchoring and skirt held a full water head with low incidence of overflow. The systems we could not dam up to maximum capacity (AQUARIWA,

INDUTAINER, FLUTSCHUTZ-DKS, Hydrobaffle) were capable of reaching higher water levels than those specified by the manufacturers. The Tiger Dam tube system was only able to achieve the protection height of 0.6 metres specified by the manufacturer by the additional use of an upstream skirt and anchoring to the ground: a test setup without skirt and anchoring threatened an early system failure. The Quick Dam Type M and Aqua Barrier systems were not available in sufficient quantities and could only be tested in combination with the AQUARIWA system. Therefore, water was only dammed up to a

height of 0.6 metres. Since the tests were carried out without any further loads caused by currents, waves, flotsam, etc., the possibility of increasing the protection heights given by the manufacturers cannot be deduced. Table 3 summarises the advantages and disadvantages of the various system types determined in the frame of our test setups.

**Table 3. Summary of the most important advantages and disadvantages of different system types.**

| **Basin system** | | |
|---|---|---|
| Advantage | - | High stability even no or small volumes of retained water (with influence of wind or similar) |
| | - | Seals well even with low volume of retained water |
| | - | Offer high safety with sand filling |
| Disadvantage | - | Installation time |
| | - | Filling material |
| **Tube system** | | |
| Advantage | - | High stability even no or small volumes of retained water (with influence of wind or similar) |
| | - | Seals well even with low volume of retained water |
| Disadvantage | - | Installation time |
| | - | Filling material |
| **Flap, trestle, dam systems** | | |
| Advantage | - | Installation time |
| | - | No filling material |
| | - | Usually overflowable |
| Disadvantage | - | Good stability only with increasing height of retained water (problematic with wind influence or similar) |
| | - | Good seal only with higher levels of retained water |

The system dismantling of the tested SBRS was generally uncomplicated. In the case of water-filled systems, it must be

ensured that the number, position and size of the openings for emptying the systems significantly influence the emptying time as well as the possibility of simple complete emptying. Even if only a small amount of residual water remains in the





system, the resulting weight can exceed a manageable level. All systems must always be cleaned and dried before being stored for reuse. The INDUTAINER system may be considered as a disposable system, as cleaning or drying is difficult owing to its intricate design. However, it has a comparatively low purchase price, so that the use of the system can be economical even if only used once. Some other SBRS also have limited disposal costs after use. This applies in particular to

systems in which the upstream skirt is (preferably) to be weighted down with sandbags. The required sandbag requirement, however, is low at approx. four sandbags per metre.

Mechanical effects, for example from flotsam or vehicle impact and vandalism, were not investigated within the scope of the test setups, but play a role in the discussion surrounding the usability of SBRS. Minor damage can be repaired with appropriate repair kits and, in the case of water- or air-filled systems, can be compensated by refilling with pumps, if

available. Furthermore, in the case of container systems and damage to individual containers with an upstream skirt, the forces can be absorbed for short times by the skirt (cf. Wagner, 2016) until it is possible to close the gap. In test series at the TU Hamburg, it was shown that SBRS can also withstand a strongish flotsam impact (cf. Flood protection, 2019; Aquaburg, 2019).

These tests, though, were carried out under idealised conditions using a bundle of wooden slats as flotsam. Since the failure

of an SBRS threatens the flooding of the hinterland with a correspondingly high damage potential and SBRS are to be regarded as more susceptible to mechanical impacts and vandalism due to their design, these aspects should be evaluated particularly critically. Mechanical effects and vandalism, though, are also relevant when using sandbag systems. In the opinion of the authors, these aspects should therefore not be an exclusion criterion, despite their particular relevance for SBRS. However, it is advisable to make higher demands on monitoring of the systems during their use.

**4 Conclusions**

Tests of various SBRS with the focus on stability, functionality and handling were carried out. The experiences from the test setups show that SBRS, owing to their functionality and their labour and time-saving characteristics as well as the lower requirement for materials, offer the potential to make operational flood defence more efficient than with the use of sandbags alone. Since SBRS are technical systems whose functional capability must be proven before they can be used, the

introduction of a test and certification system is urgently recommended. A basis for the development of a certification system according to the German standard is already available in the BWK leaflet 'Mobile Flood Protection Systems' (BWK, 2005), the international certification systems such as FM Approvals (2019) or BSI Kitemark (2019a) as well as the test results described here and in Massolle et al. (2018).

In addition to certification of the systems, other factors such as costs, local operating conditions (subsurface properties,

current, wind, waves, etc.), handling, necessary measures to minimise the risk of failure owing to mechanical defects and vandalism, sufficiently trained personnel, logistics and storage of the systems play an important role when considering the use of SBRS. Many of these aspects can be laid down in guidelines to support decision-makers with regard to the possible



use of SBRS. The cost issue can be offset by the resulting increase in the efficiency of flood protection—and thus ultimate savings through damage reduction.

**Author contribution**

Conceptualisation: B.K.; Methodology: L.L. and C.M.; Resources: C.M. and L.L.; Formal Analysis: L.L., C.M. and V.K.;
Writing—original draft preparation: L.L.; Writing—review and editing: B.K., C.M. and L.L.; Visualisation: L.L. and V.K.;
Supervision: B.K. and L.L.; Project administration: B.K.; Funding acquisition: B.K.

**Conflicts of interest**

The authors declare that they have no conflict of interest.

**Acknowledgements**

The test setups were carried out within the framework of the project 'Adaptation of flood protection training and dyke defence of the THW Training Centre Hoya to the challenges of climate change' (HWS-Bildung, duration 2016 - 2018), funded within the framework of the German adaptation strategy to climate change by the Federal Ministry for the Environment, Nature Conservation and Nuclear Safety and by the THW Foundation. We would like to thank the manufacturers and their distributors for making their systems available for the tests and our student assistants for their active
support during the test setups.

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
