# Peer review of "Sandbag Replacement Systems - Stability, Functionality and Handling"

_Natural Hazards and Earth System Sciences, 2019_

## Referee Comment (RC1) · Anonymous Referee #1 · 27 May 2019

**General comments**

The manuscript discusses the advantages and disadvantages of so-called sand-bag replacement systems (SBRS) over regular sandbags. The discussion is based on systematic tests of the SBRS with regard to their functionality, stability and handling, which are described and analyzed in Masolle at al. 2018 (https://doi.org/10.3390/geosciences8120482). One new aspect of this manuscript is the qualitative evaluation of the different SBRS in accordance with Guidelines for Loss Prevention issued by German Insurers for Mobile Flood Protection Systems and the summary of the important advantages and disadvantages of different SBRS. This discussion is certainly interesting and might also be needed at the level of municipal administrations. From a scientific point of view, however, this manuscript adds only little

what has not already been investigated and published by Masolle at al. 2018. Thus, this manuscript seems to be a summary of Masolle at al. 2018 as the stability, functionality and handling are already discussed and analyzed in this paper. The conclusions of this manuscript and the one of Masolle at al. 2018 are almost identical aswell.

Therefore, I think this manuscript could be beneficial as a summary for administrations. The scientific gain, however, seems too little to justify a publication in NHESS.

---

## Author Comment (AC1) · 7 Jun 2019

Dear Referee #1,

thank you very much for your review. Both articles – the published article Massolle et al. 2018 and the present manuscript Lankenau et al. – are based on the same testing series of sandbag and sandbag replacement systems (SBRS). Therefore, some repetitions are unavoidable for an understanding of the executed investigations like the description of the testing facility and the tested constructions (chapter 1. Introduction and chapter 2. Description of tests). Furthermore, both articles conclude that SBRS show clear benefits compared to conventional sandbag systems, but proper testing and certification is needed. Nevertheless, the research focus of the manuscript Lankenau

et al. is different from that of Massolle et al. Whereas the latter is focussing on the assessment of testing results related to barrier heights, set-up times, and seepage rates of SBRS, the present manuscript offers a broader view on functionality, handling and overall applicability of the tested SBRS. In Lankenau et al. an evaluation scheme was elaborated based on the following aspects: area of application, stability, procurement and durability, installation, dismantling and maintenance as well as logistics. These aspects are highly relevant in the development of mitigation and adaptation strategies for flood defence. Not only seepage rates are important in the planning of operational flood protection, but also questions like system applicability in a specific application area with e.g. uneven ground or required storage space for SBRS have to be answered. These results are not only highly relevant for disaster control administrations but should also be included in the development of innovative and risk-adaptive civil protection strategies. Therefore, spreading of the research results not only on administration level but also among scientists would be appreciated.

---

## Referee Comment (RC2) · Anonymous Referee #2 · 2 Sep 2019

The paper addresses a highly relevant topic: i.e. the use of alternative systems for temporary flood defence. The manuscript described a series of full scale tests. This work is relevant for a technical community, also for users (e.g. governments, consultants etc.). However, the technical and scientific novelty could be more clearly addressed (see also the comments by reviewer 1). Also, the presentation of the manuscript could be improved. I provide a number of suggestions below.

Approach:

Chapter 1 It seems that the objectives are not explicitly stated in chapter 1. Page 3 presents a lot of past studies on temporary flood defences, but a clear statement of knowledge gap and objectives seems to be missing

[Figure]

Chapter 2: I think the approach and added value could be more clearly introduced. What are typical failure modes of these systems, and which ones are you going to test? (see for a brief discussion of some failures modes also Lendering K.T., Jonkman S.N., Kok M. (2016) Effectiveness of emergency measures for flood prevention, Journal of Flood risk Management 9 (4) , 320-334.)

I would suggest to include a sketch of the basin and the test layout

I would suggest to include a table (perhaps in appendix) with some more information on the type and other properties of the systems shown in fig.2

Chapter 3: I propose to clarify which aspects (ion table 1) are based on the tests, and which are based on "manufacturers specification or "authors considerations" (p7, line 9/10)

Table 2, formulations can be more clear, e.g. "uneven ground", do you mean whether the system "can be applied on uneven ground?". Also, for the aspect of height of retained water you give a score (+,-,0) based on the retaining height. Why not just mention the retaining height in the table. Some systems may be very reliable, but "just" designed for low heads.

I would propose to include a discussion section, to outline limitations and next steps of testing, further development of these systems, certification and standardization of testing of SRS'.

Presentation: In general the use of English language could be improved, review by a native speaker would be beneficial, see detailed suggestions below.

The abstract can be improved, it could be more specific on the methods & tests, and findings & results and added value of the proposed approach

Some parts can be shortened e.g. descriptions on p5/6

Fig 3 is not clear in black and white

Examples of sentences which could be improved: • First sentence in abstract • Line 21 / 22 ("their geometry in connection with. . . . .." is not clear to me) • Etc.

---

## Author Comment (AC2) · 8 Oct 2019

During the review process it has been recommended to merge the two papers nhess-2019-164 "Sandbag Replacement Systems - Stability, Functionality and Handling" and nhess-2019-165 "Sandbagging versus Sandbag Replacement Systems: Costs, Time, Helpers, Logistics" into one paper, because both papers would benefit from this merging especially in terms of a wider introduction into the topic and the hydraulics of Sandbag Replacement Systems. The authors are very thankful for this suggestion. Therefore, the content of nhess-2019-164 was merged into nhess-2019-165, which obtained the new title "Sandbag Replacement Systems - a nonsensical and costly alternative to sandbagging?".